# Hedgehogs, Squirrels, and Blackbirds as Sentinel Hosts for Active Surveillance of *Borrelia miyamotoi* and *Borrelia burgdorferi* Complex in Urban and Rural Environments

**DOI:** 10.3390/microorganisms8121908

**Published:** 2020-11-30

**Authors:** Karolina Majerová, Václav Hönig, Michal Houda, Petr Papežík, Manoj Fonville, Hein Sprong, Natalie Rudenko, Maryna Golovchenko, Barbora Černá Bolfíková, Pavel Hulva, Daniel Růžek, Lada Hofmannová, Jan Votýpka, David Modrý

**Affiliations:** 1Department of Parasitology, Faculty of Science, Charles University, 12800 Prague, Czech Republic; jan.votypka@natur.cuni.cz; 2Institute of Parasitology, Biology Center, Czech Academy of Sciences (CAS), 37005 Ceske Budejovice, Czech Republic; honig@paru.cas.cz (V.H.); natasha@paru.cas.cz (N.R.); marina@paru.cas.cz (M.G.); ruzekd@paru.cas.cz (D.R.); modryd@vfu.cz (D.M.); 3Department of Infectious Diseases and Preventive Medicine, Veterinary Research Institute, 62100 Brno, Czech Republic; 4Department of Applied Mathematics and Informatics, Faculty of Economics, University of South Bohemia, 37005 Ceske Budejovice, Czech Republic; houda@ef.jcu.cz; 5Department of Pathology and Parasitology, Faculty of Veterinary Medicine, University of Veterinary and Pharmaceutical Sciences, 61242 Brno, Czech Republic; Reptimania@email.cz (P.P.); lada.hurkova@seznam.cz (L.H.); 6Center for Infectious Disease Control, National Institute for Public Health and the Environment, 3721 Bilthoven, The Netherlands; manoj.fonville@rivm.nl (M.F.); hein.sprong@rivm.nl (H.S.); 7Faculty of Tropical AgriSciences, Czech University of Life Sciences, 16500 Prague, Czech Republic; barbora.bolfikova@gmail.com; 8Faculty of Science, Charles University, 12800 Prague, Czech Republic; pavel.hulva@natur.cuni.cz; 9Faculty of Science, University of Ostrava, 70103 Ostrava, Czech Republic; 10Military Health Institute, Military Medical Agency, 16200 Prague, Czech Republic; 11Department of Botany and Zoology, Faculty of Science, Masaryk University, 61137 Brno, Czech Republic

**Keywords:** *Borrelia burgdorferi* sensu lato, *Borrelia miyamotoi*, European hedgehog, Northern white-breasted hedgehog, Eurasian red squirrel, Common blackbird

## Abstract

Lyme borreliosis (LB), caused by spirochetes of the *Borrelia burgdorferi* sensu lato (s.l.) complex, is one of the most common vector-borne zoonotic diseases in Europe. Knowledge about the enzootic circulation of *Borrelia* pathogens between ticks and their vertebrate hosts is epidemiologically important and enables assessment of the health risk for the human population. In our project, we focused on the following vertebrate species: European hedgehog (*Erinaceus europaeus*), Northern white-breasted hedgehog (*E. roumanicus*), Eurasian red squirrel (*Sciurus vulgaris*), and Common blackbird (*Turdus merula*). The cadavers of accidentally killed animals used in this study constitute an available source of biological material, and we have confirmed its potential for wide monitoring of *B. burgdorferi* s.l. presence and genospecies diversity in the urban environment. High infection rates (90% for *E. erinaceus*, 73% for *E. roumanicus*, 91% for *S. vulgaris*, and 68% for *T. merula*) were observed in all four target host species; mixed infections by several genospecies were detected on the level of individuals, as well as in particular tissue samples. These findings show the usefulness of multiple tissue sampling as tool for revealing the occurrence of several genospecies within one animal and the risk of missing particular *B. burgdorferi* s.l. genospecies when looking in one organ alone.

## 1. Introduction

Lyme borreliosis (LB) is one of the most abundant vector-borne diseases in Europe, with a purported annual number of human disease cases between 65,000 and 85,000 [1,2]. Only rough estimates of the LB incidence in Europe are available because the reporting systems differ among countries, and LB is not a compulsorily notifiable disease in all of them. However, the incidence numbers seem to be underestimated as results of a more recent study calculate ~232,000 new cases per year in Western Europe alone, while several countries with the highest incidence rates on the continent were not included [3,4]. The disease is caused by spirochetes of the *Borrelia burgdorferi* sensu lato (s.l.) complex [5]. Today, 22 named *B. burgdorferi* s.l. genospecies are known and well established around the world. As new species and variants are continuously being recognized, the current number of described taxa is apparently not final.

In terms of human sensitivity to *B. burgdorferi* s.l., the complex of 22 *Borrelia* genospecies can be divided into two groups:

(a) 10 genospecies with pathogenic potential that have been detected in or isolated from humans: *B. afzelii* [6], *B. bavariensis* [7], *B. bissettii* [8], *B. burgdorferi* sensu stricto (s.s.) [9], *B. garinii* [9], *B. kurtenbachii* [10], *B. lusitaniae* [11], *B. mayonii* [12], *B. spielmanii* [13], and *B. valaisiana* [14].

(b) Another 12 genospecies that have not been detected in or isolated from humans yet: *B.americana* [15], *B. andersonii* [16], *B. californiensis* [17], *B. carolinensis* [18], *B. chilensis* [19], *B. finlandensis* [20], *B. japonica* [21], *B. lanei* [22], *B. tanukii* [23], *B. turdi* [23], *B. sinica* [24], and *B. yangtzensis* [25].

In Europe, *B. burgdorferi* s.s., *B. afzelii, B. garinii*, and *B. bavariensis* are responsible for the vast majority of human disease cases [5]. Other genospecies, like *B. bissettii*, *B. spielmanii*, *B. kurtenbachii*, *B. valaisiana*, and *B. lusitaniae* may also be involved in human infections [26,27,28,29]. Humans become infected when an infectious tick feeds on their blood. In Europe, the common tick (*Ixodes ricinus*) is the most prevalent vector, ensuring the circulation of the pathogen among its natural vertebrate hosts, as well as transmission to humans [30]. Other tick species may also participate in the natural circulation of the spirochetes (e.g., *I. hexagonus*) [31,32].

Forests, preferably mixed and deciduous, are considered typical *I. ricinus* habitats, and, consequently, they are natural circulation areas for tick-borne pathogens. Nevertheless, in the last decade, attention has also been drawn to the occurrence of ticks and tick-borne pathogens in urban and peri-urban areas [33,34]. Human exposure to tick bites in urban areas may be high, thus representing a relevant health risk. People entering urban tick habitat do not pay as much attention to tick bite prevention as when visiting woodland areas; the elderly, disabled persons, and young children visit parks more frequently than woodland areas [35].

Considering the generally low mobility of ticks, birds are important hosts for tick dispersal [36] to urban habitats (i.e., gardens, city parks, cemeteries), which then provide a suitable environment for the existence of permanent tick populations. One of the important prerequisites for tick survival in urban ecosystems is the presence of vertebrate hosts suitable for feeding of all developmental stages. Synurbic populations of small rodents and birds serve as a blood-meal source for larval and nymphal ticks. Adult ticks feed on larger rodents (e.g., squirrels), insectivores (e.g., hedgehogs), or domestic pets; however, lagomorphs and roe deer may also be able to thrive in urban or peri-urban areas [34,37]. There are numerous studies reporting the occurrence of ticks in urban green spaces carrying various tick-borne human pathogens [32,33,38,39,40,41,42,43,44,45]. Nevertheless, the role of wildlife as potential reservoir hosts of tick-borne pathogens in urban areas is rarely addressed.

In our project, we focused on four vertebrate species: The European hedgehog (*Erinaceus europaeus*), the Northern white-breasted hedgehog (*E. roumanicus*), the Eurasian red squirrel (*Sciurus vulgaris*), and the Common blackbird (*Turdus merula*), which tend towards synurbization, are frequently heavily infested by ticks, and are important reservoir hosts of *B. burgdorferi* s.l. [46,47,48,49,50,51,52,53]. We used randomly found cadavers of accidentally killed animals or handicapped animals that had died in rescue centers, as they constitute an easy to collect and valuable source of biological material for tick-borne pathogen monitoring [44].

The main aims of the study were: (i) to assess the prevalence and diversity of *B. burgdorferi* s.l. genospecies in the four target species and (ii) to evaluate the potential of cadavers of accidentally killed vertebrates as a source of biological material for monitoring of zoonotic pathogens in urban environments.

## 2. Materials and Methods 

The methodology used in this study consist of cadavers sampling, dissections, tissue samples preparation, DNA isolation, multiple real-time PCR, and a set of conventional PCRs (Figure 1).

### 2.1. Hosts, Dissections, Sampling

European hedgehog (*E. europaeus*), Northern white-breasted hedgehog (*E. roumanicus*), Eurasian red squirrel (*S. vulgaris*), and Common blackbird (*T. merula*) cadavers were used as the source of biological material. All of these animals were found dead, mostly as road-kill, killed by other animals (above all by dogs and cats), or from glass collisions (birds); or died in animal rescue centers. The cadavers were kept at −20°C until further processing. Thawing of cadaver before necropsy took 4–18 h, based on the cadaver size and thawing temperature, which was 8–20 °C. After thawing, the grade of autolysis was determined (Appendix A). Only cadavers with a low (1A,1B) or intermediate (2) grade of autolysis were utilized.

The species of a given animal was identified based on characteristic morphological features [54,55,56]. However, species determination of hedgehogs based solely on morphological characteristics might be biased, especially in road-kill. Therefore, we confirmed the morphological identification using the molecular method based on mitochondrial control region analysis [57,58].

Each cadaver was weighted, its foot length was measured, and an age category was assigned, either juvenile (presence of primary dentice in mammals or juvenile feathers in birds), or adult. The gender of the animal was determined when external or internal sexual organs were present in the cadaver. The ectoparasitic infestation was recorded only as presence/absence due to the bias resulting from differences in the stretch of time from animal death to cadaver collection and the possibility of antiparasitic treatment of individuals acquired from animal rescue centers.

The animal tissues were removed under sterile conditions. Each organ was taken out with sterile surgical instruments treated with PCR Clean™ (Minerva Biolabs, Berlin, Germany) to eliminate DNA and RNA contamination from the surface. A piece of the tissue (~5 × 5 mm) was placed into a 2 mL Eppendorf tube, where it was cut into ~1-mm pieces. We collected ear tissue (or skin in the case of birds or when the ears were absent) and muscle, lung, blood, liver, spleen, urinary bladder, kidney, and brain tissue. A blood coagulum or liquid blood was obtained from the heart or thoracic cavity of the cadaver with a sterile Pasteur pipette, and a volume of 50–500 µL of blood was placed into a 2 mL Eppendorf tube. To improve the preservation of DNA and RNA (used for the detection of flaviviruses [59]), 1 mL of RLT buffer (Qiagen, Hilden, Germany) was added to each Eppendorf tube with the sample of blood or tissue and stored at −70 °C [60] until further analyses were conducted.

### 2.2. DNA Isolation

Tissue samples were homogenized in RLT buffer (Qiagen, Hilden, Germany) with the addition of beta-mercaptoethanol (10 µL of 14.3M beta-ME per 1 mL of RLT buffer) using sterile stainless-steel beads (Qiagen, Hilden, Germany) in a Tissue Lyzer II. Briefly, samples of ear, skin, muscle, lung, liver, spleen, urinary bladder, kidney, and brain tissue were prepared as 30% (*w*/*v*) suspensions. After homogenization, 20 µL of proteinase K (Qiagen, Hilden, Germany) was added, and the samples were incubated for 30 min at 57 °C. The lysate was cleared by centrifugation, and the supernatant was collected in a clean, sterile microtube. The samples of blood (10–100 µL depending on whether it was nucleated or nonnucleated blood and the sample volume obtained from a particular animal) were resuspended in 220 µL of sterile PBS, and the solution was used for DNA isolation using a DNeasy Blood and Tissue kit (Qiagen, Hilden, Germany) according to the manufacturer’s instructions. The elution volume was 200 µL.

### 2.3. Screening of the Samples through Multiplex Real-Time PCR

All PCR and real-time PCR reactions were set up in a separate area with all appropriate precautions (separated supplies, equipment, and personal safety items, pre- and post-amplification activities). The extracted DNA samples were screened for *B. burgdorferi* s.l. using multiplex real-time PCR, which also enabled the detection and identification of *B. miyamotoi*. For the detection of *B. burgdorferi* s.l., two primer pairs and two different probes were used (Appendix A), targeting outer surface protein (*ospA*) and flagellin B (*FlaB*) genes [61]. For the real-time PCR, an iQ Multiplex Powermix PCR reagent kit containing iTaq DNA polymerase (Bio-Rad Laboratories, Hercules, CA, USA) was used. The real-time PCR was performed in a LightCycler 480 Real-Time PCR System (Roche, Base, Switzerland) using the following program: an initial activation of the iTaq DNA polymerase at 95 °C for 5 min, 60 cycles of a 5 s denaturation at 95 °C, followed by a 35 s annealing-extension step at 60 °C (single point measurement at 60 °C), and a cooling cycle of 37 °C for 20 s. The analysis was performed using second derivative calculations for Cp (crossing point) values. A color compensation was conducted for the overflow of fluorescence from the dyes that were used. Curves were assessed visually in LightCycler 480 software. Samples with a positive score for one or both targets (*ospA*; *FlaB*) for *B. burgdorferi* s.l. were considered positive.

### 2.4. Conventional PCRs and Sequencing

All samples which were positive for *B. burgdorferi* s.l. according to real-time PCR were subsequently analyzed by three different conventional PCRs, targeting portions of the 5S-23S (*rrfA–rrlB*) ribosomal RNA intergenic spacer region (IGS), and genes encoding flagellin B (*FlaB*) and outer surface protein C (*ospC*), where nested PCR was performed (Appendix A). A MasterTaq kit (Eppendorf, Hamburg, Germany) containing recombinant Taq DNA polymerase from *Escherichia coli* DH1 and a 5× TaqMaster PCR enhancer was used for the amplification of the spirochete sequences. A negative control (no template) and positive control (*B. burgdorferi* sensu stricto, strain B31 DNA) was added to each amplification reaction. The PCR products were separated using gel electrophoresis, cleaned with ExoSAP-IT™ PCR Product Cleanup Reagent (Applied Biosystems™, Waltham, MA, USA), and Sanger sequencing was provided by BaseClear (Leiden, Netherlands) using internal nested PCR primers. The sequence chromatograms were checked visually and, afterwards, used for the *B. burgdorferi* s.l. genospecies identification as a comparison to sequences of known genospecies from the NCBI GenBank database.

### 2.5. Species-Specific PCR

To identify mixed infections with multiple genospecies of *B. burgdorferi* s.l. complex, the real-time PCR positive samples were screened using species-specific PCR. Primers designed to target the outer surface protein A (*ospA*) gene of *B. burgdorferi* s.s. (primer set GI), *B. garinii* (primer set GII), and *B. afzelii* (primer set GIII) (Appendix A) described by Reference [62] were used with the following modification: PCR was conducted in two steps, amplification and re-amplification. A HotStarTaq Plus Master kit (Qiagen, Hilden, Germany) was used for amplification of species-specific fragments of *ospA* genes. After the completion of primary amplification, 5 µL of the resultant mixture was transferred into a fresh PCR tube with the HotStarTaq Plus Master mixture and the same PCR primers (20 µL final volume), and re-amplification was conducted under the same conditions but only for 25 cycles. A negative control (no template) and positive control (*B. carolinensis* DNA) was added to each amplification run. The final PCR products were separated by agarose gel electrophoresis. The genospecies of *B. burgdorferi* s.l. were determined based on the difference in amplicon size. The use of the modified technique described above allows detection of the presence of multiple spirochete species in a single sample.

### 2.6. Statistical Analyses

Differences in prevalence rates were tested using contingency tables and chi-square or Fisher’s exact test. To analyze genospecies distribution among host species and tissues, we used the asymptotic (permutation-based) generalized Cochran-Mantel-Haenszel (CMH) test, as described in Reference [63], and implemented in R by the coin package [64]; the CMH statistic tests conditional independence in three-way contingency tables. As all the tests performed rejected the null hypothesis at high p-values, we complemented the analysis with post-hoc McNemar’s tests (to compare proportions in partial tables) with continuity correction and Holm adjusted *p*-values. Differences with *p* < 0.05 were considered statistically significant. 

## 3. Results

A total of 157 cadavers (60 European hedgehogs, 15 northern white-breasted hedgehogs, 22 Eurasian red squirrels, and 60 common blackbirds) were collected and dissected, resulting in a total of 862 tissue samples (Appendix A). All samples were screened for the presence of *Borrelia miyamotoi* and *B. burgdorferi* s.l. DNA by multiplex real-time PCR.

### 3.1. Prevalence of B. miyamotoi

*B. miyamotoi* DNA was detected in one or more tissues in three European hedgehog specimens (5.0%) and three Eurasian red squirrel specimens (13.6%) (Appendix A).

### 3.2. Prevalence of B. burgdorferi s.l.

The presence of *B. burgdorferi* s.l. DNA was confirmed by real-time PCR in 126 (80.2%) of the individuals (Appendix A). An individual was considered positive if the spirochete was detected in at least one of the tested tissue samples. A high proportion of positive animals was observed in all four host species (Figure 2).

### 3.3. Occurrence of Borrelia burgdorferi s.l. in Specific Tissues

Multiple tissues were sampled from each cadaver. Although some of the tissues were commonly missing, all the tissue samples available from each animal were screened for *B. burgdorferi* s.l. DNA in order to compare the detection efficiency (Table 1) and to investigate tissue tropism. When compared for all host species, ear/skin (treated as equivalent for this analysis) tissue samples were significantly more frequently positive than all of the other tested tissues (*p* < 0.01; McNemar’s chi-square test with continuity correction, Holm adjusted *p*-values), with the exception of urinary bladder tissue (Figure 3).

Similar trends were observed when tissues within the host species were compared. In a pairwise comparison, *B. burgdorferi* s.l. DNA was significantly more frequently detected in the skin (87.8% positive of 41 tested) than in other tissues of blackbirds (*p* < 0.0001). The same was true for ear tissue in European hedgehogs except for muscle and urinary bladder tissue (*p* < 0.05; McNemar’s chi-square test with continuity correction, Holm adjusted *p*-values). European hedgehog blood was significantly less infected than ear or urinary bladder tissue (*p* < 0.05; McNemar’s chi-square test with continuity correction, Holm adjusted *p*-values). In the case of squirrels, significant differences were found only between ear tissue compared to blood and kidney tissue (*p* < 0.05; McNemar’s chi-square test with continuity correction, Holm adjusted *p*-values). No significant differences were found among the tissues of Northern white-breasted hedgehogs.

### 3.4. Identification of Borrelia burgdorferi s.l. Genospecies

All real-time PCR positive samples were further investigated by four different conventional PCRs to identify the *B. burgdorferi* s.l. genospecies including mixed infections. A relatively high rate of mixed infections was found on the level of individuals (= different genospecies found in different tissues of the same animal) but also in specific tissue samples. The combinations of genospecies found in mixed infections, as well as their frequencies, are shown in Appendix A. The highest genospecies diversity was observed in the European hedgehog, either as mono-infections or as part of multiple infections. No apparent associations of genospecies were found in the mixed infections (although the relatively low total numbers prevented proper statistical analysis). Unexpectedly, *B. bissettii/carolinensis* DNA was detected in several samples (GenBank accession numbers: MW297142–297147). None of the methods allowed discrimination between these two genospecies as the DNA was found only in samples with mixed infections, and short *ospA* sequences obtained from species-specific PCR were not sufficient for distinguishing due to the close genetic relatedness of these two *Borrelia* genospecies.

The proportional representation of the genospecies differed significantly among the hosts (Figure 4) (Asymptotic General Independence Test *p* < 0.0001). Particularly in the case of European hedgehogs, *B. afzelii* was significantly more frequent than any other genospecies (*p* < 0.0001; McNemar’s chi-square test with continuity correction, Holm adjusted *p*-values), and *B. bissetii/carolinensis* and *B. valaisiana* were less frequent than all but *B. bavariensis* and *B. garinii* (*p* < 0.05; McNemar’s chi-square test with continuity correction, Holm adjusted *p*-values). No significant differences were found in a pairwise comparison of the genospecies infecting the Northern white-breasted hedgehog. *B. burgdorferi* s.s. was significantly more frequent in squirrels than *B. bissettii/carolinensis* (*p* < 0.05; McNemar’s chi-square test with continuity correction, Holm adjusted *p*-values), and *B. bavariensis* was significantly less frequently found in blackbirds than *B. garinii* and *B. valaisiana* (*p* < 0.05; McNemar’s chi-square test with continuity correction, Holm adjusted *p*-values). 

### 3.5. Comparison of Occurrence of Borrelia burgdorferi s.l. Genospecies in Investigated Tissues

The overall distribution of *B. afzelii*, *B. garinii*, and *B. valaisiana* among the tissues differs significantly (Asymptotic General Independence Test, *p* < 0.0001) in general analysis ignoring the host species (Table 2). In a pairwise comparison, *B. afzelii* DNA was significantly more frequently found in ear/skin and muscle tissues than in all other tissues except urinary bladder and kidney tissue (*p* < 0.05, McNemar’s chi-square test with continuity correction, Holm adjusted *p*-values). In the case of *B. garinii,* ear/skin tissue was significantly more frequently infected than blood and brain tissue (*p* < 0.05, McNemar’s chi-square test). In the case of *B. valaisiana,* ear/skin tissue was significantly more infected than blood, liver, and muscle tissue (*p* < 0.05, McNemar’s chi-square test).

Distribution of *B. burgdorferi* s.l. genospecies also differs among the tissues within each of the four host species (Table 3). In a pairwise comparison (McNemar´s test), significant differences were found only for European hedgehog ear and muscle tissue and Common blackbird skin tissue. In European hedgehog ear tissue, *B. afzelii* was significantly more frequently detected than any other species except for *B. spielmanii* and *B. bavariensis* (*p* < 0.01, McNemar’s chi-square test with continuity correction, Holm adjusted *p*-values). The same genospecies were also more frequent in European hedgehog muscle tissue than all other genospecies except for *B. bavariensis*, *B. burgdorferi* s.s., and *B. bissettii/carolinensis* (*p* < 0.01, McNemar’s chi-square test with continuity correction, Holm adjusted p-values). In Common blackbird skin tissue, *B. garinii* and *B. valaisiana* were significantly more frequent than all other genospecies except for *B. afzelii* (*p* < 0.01, McNemar’s chi-square test with continuity correction, Holm adjusted *p*-values).

## 4. Discussion

*Borrelia miyamotoi*, a causative agent of human nonspecific febrile illness, has been previously detected in various host species, mostly rodents, but also other vertebrates, e.g., passerine birds and hedgehogs (reviewed in Reference [65]). Although our study was primarily focused on *B. burgdorferi* s.l., using a multiplex real-time PCR protocol that includes *B. miyamotoi* specific primers and probe, we detected the DNA of this *Borrelia* in a few European hedgehog and Eurasian red squirrel individuals. These findings support the previous reports on *B. myiamotoi* in Europe [66,67,68] and suggest that squirrels and hedgehogs contribute to the natural transmission cycle of this pathogen.

Although three vertebrate species chosen for this study (*E. europaeus*, *S. vulgaris*, *T. merula*) have been listed as reservoir hosts of *B. burgdorferi* s.l. for a long time [32,49,50], and European hedgehog was reported as a potential reservoir host [52], there is only a limited number of studies focused on the prevalence of *B. burgdorferi* s.l. or genospecies diversity in these hosts. Moreover, most of the studies (especially in the case of blackbirds) investigated the attached ticks not the host blood or tissues. Using cadavers as a source of biological material of different tissues enabled us to detect various genospecies of *B. burgdorferi* s.l. in all target host species and gain insight into prevalence, tissue specificity patterns and mixed infections.

We detected DNA of four different genospecies in Eurasian red squirrels: *B. burgdorferi* s.s., *B. garinii*, *B. afzelii*, and, for the first time, also *B. bissettii/carolinensis* in one specimen. *B. burgdorferi* s.s., *B. afzelii*, and *B. garinii* have been repeatedly observed in Eurasian red squirrels [49,69,70,71,72], and *Borrelia* genospecies mixed infections in analyses based on skin or ear tissue samples have also been reported previously [49,70]. In comparison to most of the published studies, we observed a high prevalence (91%), similar to results from Norway (88%) where a real-time PCR detection method and a similar number of tested animals were used [72]. *B. burgdorferi* s.l. DNA was most frequently found in ear/skin tissue (with significant differences compared to blood and kidney tissue). However, screening of multiple host tissues revealed the presence of mixed infections, and, among them, DNA of *B. bissettii/carolinensis* was also detected.

The European hedgehog is a well-recognized reservoir host for several *Borrelia* genospecies: *B. afzelii*, *B. spielmanii*, *B. bavariensis*, *B. garinii*, and *B. burgdorferi* s.s. have been repeatedly reported [32,52,73,74,75], and the role of European hedgehog in *Borrelia* transmission has also been proven experimentally [32]. The vast majority of European hedgehog specimens tested in our study (90%) were found to be positive for at least one of the previously reported *Borrelia* genospecies (the infection rate was higher when compared to other studies). DNA of *B. valaisiana* and *B. bissettii/carolinensis* was detected for the first time in the tissues of these insectivores. While *Borrelia* genospecies mixed infections have been observed in European hedgehogs before [32,52], we detected the co-occurrence of up to five genospecies in particular individuals. Our results strongly support the hypothesis that European hedgehog plays an important role in the maintenance of various *Borrelia* genospecies in natural cycles, including the urban environment.

Of all of the target host species, data about *Borrelia* infections in the Northern white-breasted hedgehog are the scarcest. Despite the fact that this hedgehog species seems to be an important host of the nidicolous hedgehog tick, *I*. *hexagonus*, as well as *I*. *ricinus*, its role in harboring zoonotic agents has rarely been examined [37,52,76,77]. According to our knowledge, there have only been two *B. burgdorferi* s.l. genospecies described from Northern white-breasted hedgehog to date, *B*. *afzelii* and *B*. *bavariensis* [44,52,76]. In our study, more than 70% of the tested specimens were *Borrelia*-positive, and, interestingly, mixed infections of two or three genospecies were recorded considerably often, thanks to the parallel analysis of several tissue samples per individual. We detected a total of three genospecies: *B. afzelii*, *B. burgdorferi* s.s., and *B. garinii*; to our knowledge, this was the first detection of two of them, namely *B. burgdorferi* s.s. and *B. garinii*, in this vertebrate species.

Birds, especially passerine birds, are known as important hosts of zoonotic tick-borne pathogens, which they can transport across geographical barriers (e.g., Reference [78]). The potential of Common blackbirds to serve as reservoir hosts for *B. burgdorferi* s.l. was proven more than twenty years ago [50]. Since that time, several studies have been performed to describe *Borrelia* genospecies diversity and prevalence in this avian species, and three of them, *B. garinii*, *B. valaisiana*, and *B. turdi*, are considered to be more closely associated with the Common blackbird [53,78,79,80]. In our set of tested Common blackbirds, *B. garinii* and *B. valaisiana* were found most frequently, in line with our expectations. *B*. *bavariensis*, a rodent-associated genospecies closely related to *B. garinii* [81], was found in a single individual. In six (10%) Common blackbird individuals, DNA of *B. afzelii* was detected in the skin or muscle tissue. These findings can contribute to the discussion about its strict mammal-host specificity. There are several reports of *B. afzelii* detected in ticks attached to birds (including Common blackbirds), in the vast majority nymphs, where other means of infection may play a role [79,82], but also in tick larvae [83]. Since transovarial transmission of *B. burgdorferi* s.l. from female ticks to larvae has not been proven to date, that and transmission among co-feeding ticks seems not to be the most likely explanation [82,83], while the hypothesis of birds serving as possible hosts for this *Borrelia* genospecies is still under discussion [83]. On the other hand, a recent experimental infection study concluded that avian blood most likely has a borreliacidal effect on *B. afzelii* [82,84]. According to our observations, skin tissue analysis seems to be the most suitable method for *Borrelia* infection testing in birds, supporting the results of a recent study [85] proving the dermatropism of *B. garinii* and *B. valaisiana* in common blackbirds.

Overall, the use of cadavers (and examination of multiple tissue samples) of all target host species seems to be one of the most suitable approaches for studying the prevalence and *Borrelia* genospecies diversity in urban and periurban areas. We were able to detect the vast majority of *Borrelia* genospecies previously described in the given host species. A considerably high proportion of real-time PCR positive samples could not be amplified in conventional PCRs or (more often) poor sequences were obtained, which reduced the genospecies identification success rate. Nevertheless, conventional species-specific PCR enabled us to detect some of the previously unidentified genospecies, and, furthermore, we were able to distinguish mixed infections with multiple genospecies in many cases. Successful genospecies identification has shown to be difficult also in *Borrelia* focused studies in the past, and we suppose that the higher sensitivity of real-time PCR assay compared to conventional PCRs is the most likely explanation for this phenomenon [72,75].

## 5. Conclusions

A total of 157 cadavers (60 European hedgehogs, 15 Northern white-breasted hedgehogs, 22 Eurasian red squirrels, and 60 Common blackbirds) were collected and dissected resulting in a total of 862 tissue samples. All of the samples were screened for the presence of *Borrelia miyamotoi* and *B. burgdorferi* s.l. by multiplex real-time PCR. Several individuals (3 specimens of *E. europaeus* (5%) and 3 specimens of *S. vulgaris* (14%)) were found to be *B. miyamotoi*-positive in one or multiple tissues. High infection rates (90% for *E. europaeus*, 73% for *E. roumanicus*, 91% for *S. vulgaris*, and 68% for *T. merula*) were observed in all four target host species for *B. burgdorferi* s.l. The methodological approach using DNA from different tissue samples and a set of conventional PCRs enabled us to detect a wide spectrum of various *B. burgdorferi* s.l. genospecies in all four vertebrate hosts and revealed a considerably high occurrence of mixed infections on the level of individuals, as well as in particular tissues. According to our knowledge, we are the first to report the detection of *B. bissettii/carolinensis* DNA in tissue samples of Eurasian red squirrels and European hedgehogs, *B*. *burgdorferi* s.s. and *B*. *garinii* in Northern white-breasted hedgehogs, *B. valaisiana* in European hedgehogs, and *B. bavariensis*, as well as *B. afzelii*, in Common blackbirds. We conclude that all of the four target vertebrate species contribute to the maintenance of zoonotic *Borrelia* species, and the cadavers of accidentally killed vertebrates are a valuable source of biological material for monitoring of *B. burgdorferi* s.l. presence and diversity.

## Figures and Tables

**Figure 1 microorganisms-08-01908-f001:**
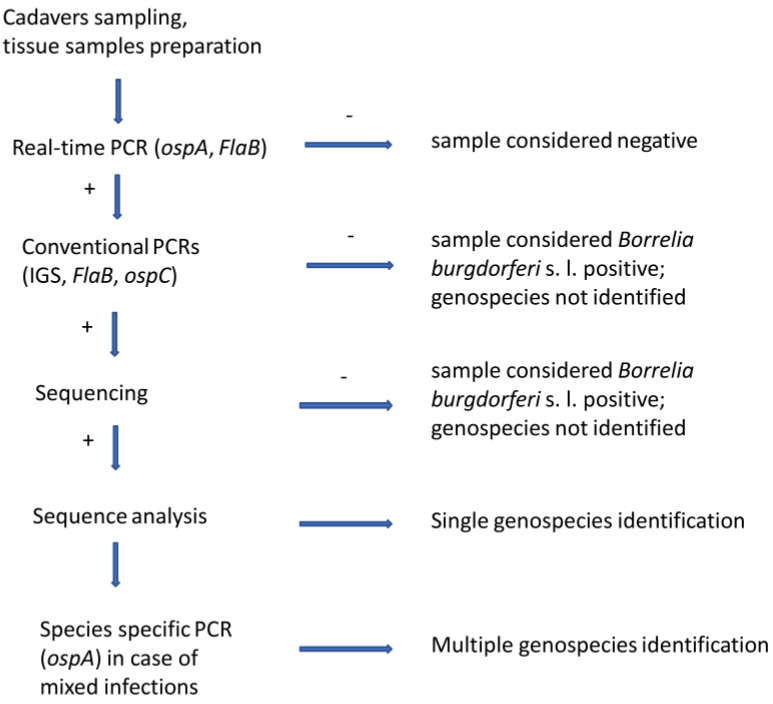
Methodology scheme. + means positive results; − means negative results.

**Figure 2 microorganisms-08-01908-f002:**
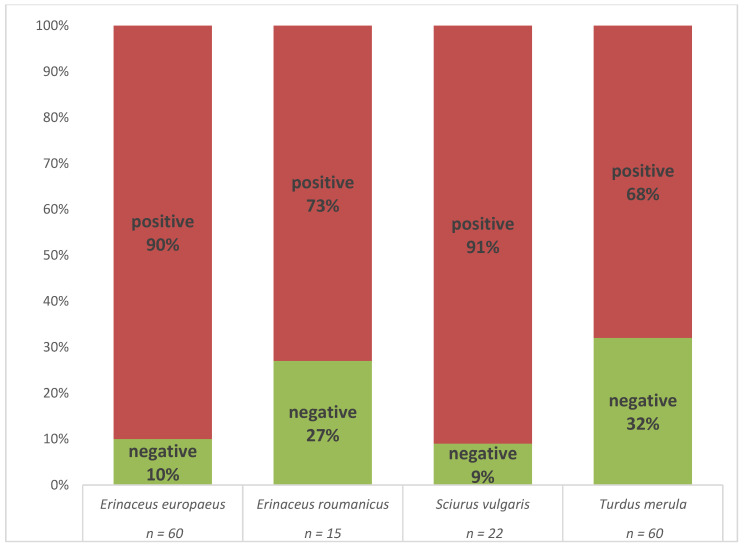
Comparison of *B. burgdorferi* s.l. prevalence in four host species as assessed by multiplex real-time PCR.

**Figure 3 microorganisms-08-01908-f003:**
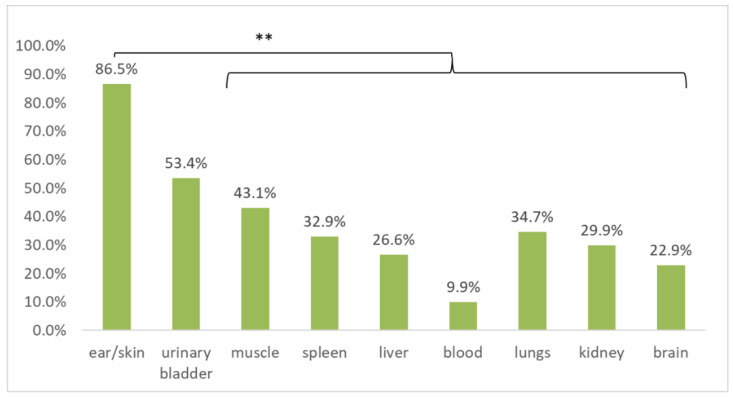
Comparison of *B. burgdorferi* sensu lato prevalence as assessed by multiplex real-time PCR in different tissue samples collected in cadavers of European hedgehogs, Northern white-breasted hedgehogs, Eurasian red squirrels, and Common blackbirds. ** indicates statistically significant differences (*p* < 0.01; McNemar’s chi-square test with continuity correction, Holm adjusted *p*-values).

**Figure 4 microorganisms-08-01908-f004:**
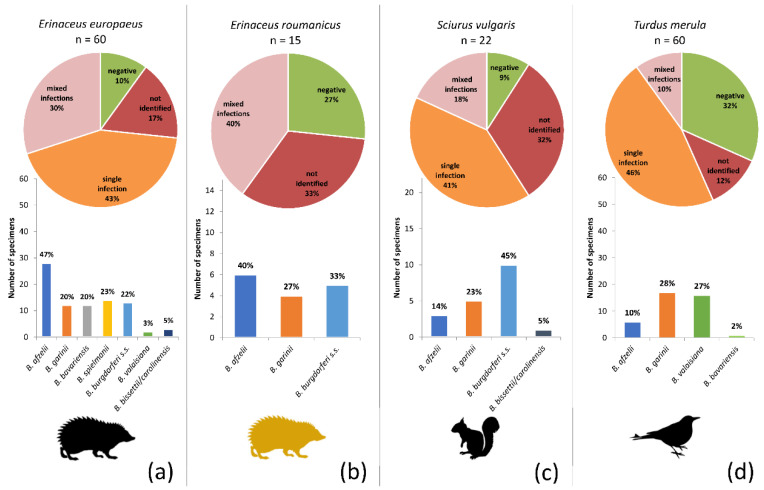
Prevalence of *Borrelia burgdorferi* s.l. genospecies among the four target host species: (**a**) European hedgehog, (**b**) Northern white-breasted hedgehog, (**c**) Eurasian red squirrel, and (**d**) Common blackbird. The pie charts indicate the proportions of individuals not infected (green), infected by a single genospecies (orange), infected by multiple genospecies (pink) and those samples that were positive in *B. burgdorferi* s.l. real-time PCR, but in which genospecies identification was not possible (dark red). The bar charts show the proportions of the genospecies as found by a combination of sequencing and species-specific PCR (summed for single and multiple infections).

**Table 1 microorganisms-08-01908-t001:** Comparison of different tissues for the detection efficiency of *Borrelia burgdorferi* s.l. DNA by real-time PCR. The percentage represents the portion of positive samples of a particular tissue from the total number of *B. burgdorferi* s.l. positive individuals (with the tissue available).

Host Species (Number of Individuals)	Ear/Skin *	Muscle	Blood	Lungs	Liver	Spleen	Urinary Bladder	Kidney	Brain
*Erinaceus europaeus* (*n* = 54)	87.0%	63.5%	17.8%	38.3%	44.4%	40.0%	60.9%	40.0%	33.3%
*E. roumanicus* (*n* = 11)	72.7%	54.5%	12.5%	22.2%	22.2%	20.0%	45.5%	22.2%	37.5%
*Sciurus vulgaris* (*n* = 20)	90.0%	35.0%	5.9%	31.6%	15.8%	22.2%	37.5%	5.6%	7.1%
*Turdus merula* (*n* = 41)	87.8%	17.5%	0.0%	x	11.1%	x	x	x	13.2%

* For mammalian hosts, ear tissue was sampled; for birds, skin from the head was obtained; *n*: number of *B. burgdorferi* s.l. positive individuals; x: tissue not sampled.

**Table 2 microorganisms-08-01908-t002:** Distribution of *B. burgdorferi* sensu lato genospecies among the tissues of European hedgehog, Northern white-breasted hedgehog, Eurasian red squirrel, and Common blackbird.

	Ear/Skin	Muscle	Blood	Lungs	Liver	Spleen	Urinary Bladder	Kidney	Brain
*B. afzelii* ***	27.84%	23.71%	2.06%	4.12%	6.19%	7.22%	15.46%	7.22%	6.19%
*B*. *garinii****	39.58%	20.83%	0.00%	6.25%	8.33%	2.08%	16.67%	2.08%	4.17%
*B. bavariensis*	31.58%	5.26%	5.26%	10.53%	10.53%	5.26%	21.05%	5.26%	5.26%
*B. burgdorferi* s.s.	12.73%	12.73%	10.91%	16.36%	5.45%	7.27%	18.18%	5.45%	10.91%
*B. spielmanii*	26.09%	30.43%	0.00%	4.35%	4.35%	8.70%	13.04%	13.04%	0.00%
*B. valaisiana* ***	83.33%	5.56%	0.00%	0.00%	0.00%	0.00%	0.00%	0.00%	11.11%
*B.* *bissettii/carolinensis*	16.67%	0.00%	16.67%	16.67%	16.67%	33.33%	0.00%	0.00%	0.00%

The percentage indicates the portion of genospecies found in the tissue (the darker the color, the higher prevalence of the particular genospecies in a particular tissue). *** indicates statistically significant differences in the distribution of *B. burgdorferi* s.l. genospecies among different tissues (*p* < 0.001; Asymptotic General Independence Test).

**Table 3 microorganisms-08-01908-t003:** Distribution of *B*. *burgdorferi* sensu lato genospecies among the different tissues in European hedgehog (*E. europaeus*), Northern white-breasted hedgehog (*E. roumanicus*), Eurasian red squirrel (*S. vulgaris*), and Common blackbird (*T. merula*).

*E. europaeus*	Total	Ear	Muscle	Blood	Lungs	Liver	Spleen	Urinary Bladder	Kidney	Brain
*B. afzelii*	74	25.68%	22.97%	1.35%	4.05%	8.11%	6.76%	14.86%	8.11%	8.11%
*B. garinii*	16	12.50%	31.25%	0.00%	0.00%	18.75%	0.00%	31.25%	6.25%	0.00%
*B. bavariensis*	18	27.78%	5.56%	5.56%	11.11%	11.11%	5.56%	22.22%	5.56%	5.56%
*B. burgdorferi* s.s.	23	4.35%	4.35%	17.39%	21.74%	4.35%	8.70%	21.74%	4.35%	13.04%
*B. spielmanii*	23	26.09%	30.43%	0.00%	4.35%	4.35%	8.70%	13.04%	13.04%	0.00%
*B. valaisiana*	2	0.00%	50.00%	0.00%	0.00%	0.00%	0.00%	0.00%	0.00%	50.00%
*B. bissettii/carolinensis*	5	20.00%	0.00%	20.00%	20.00%	0.00%	40.00%	0.00%	0.00%	0.00%
***E. roumanicus***	**Total**	**Ear**	**Muscle**	**Blood**	**Lungs**	**Liver**	**Spleen**	**Urinary Bladder**	**Kidney**	**Brain**
*B. afzelii*	12	16.67%	33.33%	0.00%	0.00%	0.00%	16.67%	25.00%	8.33%	0.00%
*B. garinii*	4	0.00%	50.00%	0.00%	0.00%	0.00%	0.00%	50.00%	0.00%	0.00%
*B. bavariensis*	0	0.00%	0.00%	0.00%	0.00%	0.00%	0.00%	0.00%	0.00%	0.00%
*B. burgdorferi* s.s.	16	0.00%	4.12%	1.03%	2.06%	1.03%	2.06%	3.09%	1.03%	2.06%
*B. spielmanii*	0	0.00%	0.00%	0.00%	0.00%	0.00%	0.00%	0.00%	0.00%	0.00%
*B. valaisiana*	0	0.00%	0.00%	0.00%	0.00%	0.00%	0.00%	0.00%	0.00%	0.00%
*B. bissettii/carolinensis*	0	0.00%	0.00%	0.00%	0.00%	0.00%	0.00%	0.00%	0.00%	0.00%
***S. vulgaris***	**Total**	**Ear**	**Muscle**	**Blood**	**Lungs**	**Liver**	**Spleen**	**Urinary Bladder**	**Kidney**	**Brain**
*B. afzelii*	5	40.00%	0.00%	20.00%	20.00%	0.00%	0.00%	20.00%	0.00%	0.00%
*B. garinii*	9	22.22%	11.11%	0.00%	33.33%	11.11%	11.11%	11.11%	0.00%	0.00%
*B. bavariensis*	0	0.00%	0.00%	0.00%	0.00%	0.00%	0.00%	0.00%	0.00%	0.00%
*B. burgdorferi* s.s.	16	37.50%	12.50%	6.25%	12.50%	6.25%	0.00%	12.50%	6.25%	6.25%
*B. spielmanii*	0	0.00%	0.00%	0.00%	0.00%	0.00%	0.00%	0.00%	0.00%	0.00%
*B. valaisiana*	0	0.00%	0.00%	0.00%	0.00%	0.00%	0.00%	0.00%	0.00%	0.00%
*B. bissettii/carolinensis*	1	0.00%	0.00%	0.00%	0.00%	100.00%	0.00%	0.00%	0.00%	0.00%
***T. merula***	**Total**	**Skin**	**Muscle**	**Blood**		**Liver**				**Brain**
*B. afzelii*	6	66.67%	33.33%	0.00%		0.00%				0.00%
*B. garinii*	19	78.95%	10.53%	0.00%		0.00%				10.53%
*B. bavariensis*	1	100.00%	0.00%	0.00%		0.00%				0.00%
*B. burgdorferi* s.s.	0	0.00%	0.00%	0.00%		0.00%				0.00%
*B. spielmanii*	0	0.00%	0.00%	0.00%		0.00%				0.00%
*B. valaisiana*	16	93.75%	0.00%	0.00%		0.00%				6.25%
*B. bissettii/carolinensis*	0	0.00%	0.00%	0.00%		0.00%				0.00%

The percentage indicates the portion of genospecies found in the tissue (the darker the color, the higher prevalence of the particular genospecies in a particular tissue).

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
