# Peer review of "Hedgehogs, Squirrels, and Blackbirds as Sentinel Hosts for Active Surveillance of Borrelia miyamotoi and Borrelia burgdorferi Complex in Urban and Rural Environments"

_microorganisms, 2020, doi:10.3390/microorganisms8121908_

Round 1
Reviewer 1 Report
This is an interesting and informative study that deserves to be published. I would, however, like to see the methodology expanded concerning PCR. Are the primer sequences previously published? If so, please provide references that include primer sequences. If they have not been published it would be useful to provide primer sequences -- perhaps in a table. If they are proprietary this should be indicated.
Author Response
Dear reviewer,
thank you very much for your comments and suggestions!
All of the primers were previously published and their sequences (as well as detailed protocols for PCR runs) are listed in Supplementary table 2 (Table S2). By the mistake (during change of references format), the references were not included in Table S2 and we highly appreciated that you noticed that! For now, we added them to the Table S2 and the references which were missing in the reference list of the manuscript, were added there as well.
The changes we made you can see in the attachment.
Thank you one more time for your attentive reading of the manuscript and helping us to avoid such a mistake!
Sincerly,
Karolina Majerová

Reviewer 2 Report
The reviewer wonders whether 'co-infection' is the optimal term to describe simultaneous infections by multiple strains of the same species. Usually, in the reviewers experience, the term 'co-infection' is used to denote infections of multiple species within the same host. One might use 'co-infection' with B. miyamotoi since, apparently, many consider this within the relapsing fever 'clade' (if that is the correct designation). So, it is a 'small' point, but the reviewer wonders if some other term or phrase aside from 'co-infection' such as 'infection by multiple strains of borreliae' - perhaps that is too cumbersome a phrase. Again, usually the term 'co-infection' in the setting of a borrelial infection would refer to babesia piroplasms, or anaplasma or ehrlichia. Again, a 'small point' but readers might be confused by the term 'co-infection' when referring merely to a variety of strains of Bbsl.
Regarding wording, again, a small point, but on line 39 there is some unnecessary redundancy: "...when looking in only one organ alone." might be sufficient to say "...in only one organ." or "...in one organ alone."
Very interesting and detailed study that obviously entailed a lot of work. this type of approach might also be utilized in the study of human tissues that become available through either surgery, biopsy or autopsy. It might be worth a comment in that regard on your mss. although that is beyond the scope of your attention. Still, the diversity of borreliae in your cadaveric animal study and the varying distributions of strains in various tissues and organs cannot help but raise analogous questions in the study of human cases of borreliosis.
Author Response
Dear reviewer,
thank you very much for your comments and suggestions!
Our responses are divided to particular comments you mentioned:
Point 1: The reviewer wonders whether 'co-infection' is the optimal term to describe simultaneous infections by multiple strains of the same species. Usually, in the reviewers experience, the term 'co-infection' is used to denote infections of multiple species within the same host. One might use 'co-infection' with B. miyamotoi since, apparently, many consider this within the relapsing fever 'clade' (if that is the correct designation). So, it is a 'small' point, but the reviewer wonders if some other term or phrase aside from 'co-infection' such as 'infection by multiple strains of borreliae' - perhaps that is too cumbersome a phrase. Again, usually the term 'co-infection' in the setting of a borrelial infection would refer to babesia piroplasms, or anaplasma or ehrlichia. Again, a 'small point' but readers might be confused by the term 'co-infection' when referring merely to a variety of strains of Bbsl.
Response 1: Thank you for this comment! We have discussed the term „co-infections“ with the co-authors before and we also were not sure. We have previously considering the term „mixed infections“. Thanks to your suggestion we were looking for the definitions and found this citation in the Research Gate discussion from Dr. Hannah Tanner: „In clinical bacteriology what we usually mean by a mixed infection is where a single infection is caused by a variety of bacterial species which are simultaneous causing the same infection. For example: peritonitis cased by all kinds of different gut bacteria and yeasts. Co-infection is more likely to mean the person has got two separate infections going on at the same time (like HIV and TB).“ On the base of this, we replaced all the „co-infections“ by „mixed infections“ on the lines 35, 174, 245, 246, 248, 251, 324, 328, 333, 342, 384, as well as the term „mix infection“ by „mixed infections“ in Figure 4 to have it unified. Hope it will not be confusing any more.
Point 2: Regarding wording, again, a small point, but on line 39 there is some unnecessary redundancy: "...when looking in only one organ alone." might be sufficient to say "...in only one organ." or "...in one organ alone."
Response 2: The sentence was changed on the line 39 and the term „when looking in one organ alone“ was used. Thank you for your attentive reading of the manuscript!
Point 3: Very interesting and detailed study that obviously entailed a lot of work. this type of approach might also be utilized in the study of human tissues that become available through either surgery, biopsy or autopsy. It might be worth a comment in that regard on your mss. although that is beyond the scope of your attention. Still, the diversity of borreliae in your cadaveric animal study and the varying distributions of strains in various tissues and organs cannot help but raise analogous questions in the study of human cases of borreliosis.
Response 3: We really appreciate your nice words and the potential of our approach and results to be used in clinical perspective – human patients! However, the problematic of using the surgery/biopsy/autopsy tissue samples for human cases of borreliosis is really extended the scope of this manuscript and we don´t feel to be qualified for such recommendation. Thank you in advance for understanding!
Thank you one more time for your attentive reading of the manuscript and helping us to avoid mistakes!
Sincerely,
Karolina Majerová
Reviewer 3 Report
Revision of manuscript microorganisms-1020197
Dear Authors,
Your manuscript entitled “Hedgehogs, squirrels and blackbirds as sentinel hosts for active surveillance of Borrelia miyamotoi and Borrelia burgdorferi complex in urban and rural environments” is a very interesting big work on the presence of various Borrelia genospecies in 3 different small mammals and Turdus merula. The work is well planned and conducted. Results are well presented and discussed. Obtained data contribute to the knowledge about this important zoonosis.
I have no comment or corrections for Authors only two little suggestion.
- Introduction:
- Lines 59-62: genospecies name in italics
- Materials and Methods.
- Figure 1: In my opinion, this figure is very useful, because it clearly show and resume the investigation workflow, but it is not cited in the text and in my opinion 2 lines of introduction and explanation could be necessary; furthermore, starting the material and methods section with this figure is not very beautiful, only by an esthetic point of view.
- .
Sincerely
The Reviewer
Author Response
Dear reviewer,
thank you very much for your comments and suggestions!
We changed the manuscript according to your suggestions:
Point 1: Introduction: Lines 59-62: genospecies name in italics
Response 1: Thank you for your attentive reading of the manuscript! Of course, the italics in genospecies names disappeared by the mistake in this paragraph! Now corrected in the lines 59-62. Thank you!
Point 2: Materials and Methods. Figure 1: In my opinion, this figure is very useful, because it clearly show and resume the investigation workflow, but it is not cited in the text and in my opinion 2 lines of introduction and explanation could be necessary; furthermore, starting the material and methods section with this figure is not very beautiful, only by an esthetic point of view.
Response 2: Thank you for your suggestion! We added the introduction sentence on the lines 98-99 „The methodology used in this study consist of cadavers sampling, dissections, tissue samples preparation, DNA isolation, multiple real-time PCR and a set of the conventional PCRs (Figure 1).“
Thank you one more time for your review!
Sincerely,
Karolina Majerová